# Preoperative binaural beats reduce remimazolam dosage and enhance safety in anesthesia induction: A randomized controlled trial

Hyun-Chang Kim[1], Jin Young Sohn[2], Myoung Hwa Kim[1], Yoon Jung Kim[2], Chul Ho Chang[1], Jeong-Hwa Seo[2]*

1 Department of Anesthesiology and Pain Medicine and Anesthesia and Pain Research Institute, Yonsei University College of Medicine, Seoul, Republic of Korea, 2 Department of Anesthesiology and Pain Medicine, Seoul National University Hospital, Seoul National University College of Medicine, Seoul, Republic of Korea

* eong77@snu.ac.kr

## Abstract

Binaural beats, a form of auditory stimulation, are thought to reduce anxiety and anesthetic requirements through brainwave entrainment. Remimazolam offers advantages in terms of rapid onset and offset of action and hemodynamic stability. However, the optimal remimazolam dose for anesthesia induction remains unclear and there are concerns regarding variability in response and potential side effects at higher doses. This study investigated the effects of preoperative binaural beats on the remimazolam dose required for loss of consciousness during general anesthesia induction. In this randomized, prospective, single center study, 72 patients undergoing general anesthesia were allocated to two groups: the binaural sound (B group) or the control group. The B group listened to binaural sounds (1-Hz frequency difference) for 30 min preoperatively, while the control group did not. The B group required a significantly lower remimazolam dose for loss of consciousness ($15.0 \pm 3.6$ vs. $17.7 \pm 4.5$ mg, p = 0.006) and achieved loss of consciousness faster ($140 \pm 29$ vs. $168 \pm 47$ s, p = 0.003) than the control group. The incidence of hypotension was lower in the B group than in the control group (6 vs. 28%, p = 0.024). Electroencephalography spectral analysis revealed no significant between-group differences. Binaural beats significantly reduced the remimazolam dose required for loss of consciousness and shortened the time to loss of consciousness, while reducing the incidence of hypotension during anesthesia induction. Binaural beats are an effective, non-invasive method of enhancing efficiency and safety in anesthesia induction when using remimazolam infusion.

## Trial registration

ClinicalTrials.gov NCT06099977

**Data availability statement:** The datasets generated and/or analyzed during the current study are available in Mendeley Data repository, https://data.mendeley.com/datasets/swxgj79jpy/1.

**Funding:** This work was supported by the Department of Anesthesiology and Pain Medicine and Anesthesia and Pain Research Institute, Yonsei University College of Medicine. This research was supported by a special research grant funded by the Korean Society of Neuroscience in Anesthesiology and Critical Care (KSNACC-2024) and a new faculty research seed money grant from the Yonsei University College of Medicine for 2024 (2024-32-0075).

**Competing interests:** The authors have declared that no competing interests exist.

## Introduction

The quest for safer and more efficient methods of anesthesia induction remains a challenge. Anesthesia induction requires precise dosing of anesthetic agents to achieve optimal sedation, while minimizing the risks associated with higher doses. Remimazolam, a novel benzodiazepine derivative, is a promising candidate because of its rapid onset and offset of action, offering potential advantages in terms of hemodynamic stability and reduced respiratory depression [1–3]. However, the optimal remimazolam dose for anesthesia induction is still unclear, with concerns regarding individual variability in response and potential side effects at higher doses [4–7].

Binaural beats, a form of auditory stimulation involving slightly different frequencies being presented to each ear, may reduce anxiety and anesthetic requirements, potentially through brainwave entrainment or psychological relaxation [8–10]. While some evidence suggests binaural beats engage the brainstem's superior olivary complex to produce a coherent neural response [11], their exact mechanism, whether through neural entrainment or anxiety reduction, remains unclear due to the limited number of direct comparative studies [12].

This study investigates the effects of preoperative binaural beats on the remimazolam dose required for loss of consciousness (LoC) during general anesthesia induction, to explore their potential to optimize sedation and to determine their underlying mechanisms, such as anxiety reduction or neural entrainment.

We also aimed to elucidate whether the incorporation of binaural beats into the preinduction phase could optimize the sedative properties of remimazolam, thereby reducing the required dosage and associated risks.

## Materials and methods

This randomized, prospective, single-center, two-arm study was approved by the Investigative Review Board of Yonsei University Gangnam Severance Hospital in Seoul, Korea (document number: 2023-0759-001) on October 6, 2023. The study was registered at ClinicalTrials.gov (NCT06099977; November 1, 2023). Written informed consent was obtained prior to patient enrolment.

### Participants

Patients with an American Society of Anesthesiologists (ASA) physical status of 1–2, aged 20–60 years, with an ideal bodyweight of 50–80 kg, and scheduled for general anesthesia in November 1 and December 15, 2023 were included. The ideal bodyweight was calculated as follows: for men, 50 + 0.91 × (height in cm − 152.4), and for women, 45.5 + 0.91 × (height in cm − 152.4) [13]. Patients were excluded if they had a hearing disability; had used opioids or sedatives within the past week; were dependent on alcohol or drugs; had hypersensitivity to remimazolam; or had arrhythmia, cardiovascular disease, heart failure, hypovolemia, or liver failure.

### Randomization and intervention

Seventy-two patients were randomly allocated to two groups (the binaural sound [B group] or no sound group [control group]) in a 1:1 ratio based on a

computer-generated randomization list, which was placed in a sealed opaque envelope. Patients in the B group used headphones to listen to real-time binaural sounds for 30 min in the anesthesia pretreatment room. Real-time binaural sounds with a frequency difference of 1 Hz (431 Hz on the left side and 432 Hz on the right) were used. The study adhered to the approved protocol (version 1.1), with a binaural beats frequency difference of 1 Hz selected within the approved range of 1–4 Hz. The binaural beat frequency difference was set at 1 Hz to target very low-frequency neural oscillations associated with reduced levels of consciousness. Previous studies have demonstrated that low-frequency binaural beats (< 1 Hz) can facilitate transitions toward sleep or slow-wave activity, suggesting their potential to modulate baseline arousal states prior to anesthesia induction [14,15]. In addition, slow cortical oscillations around 1 Hz are a characteristic feature of unconscious states during general anesthesia, supporting the physiological relevance of this frequency range [16]. Based on this evidence, a 1-Hz frequency difference was selected as a theoretically plausible stimulus to promote a lower arousal state before anesthetic induction. Patients in the control group used headphones but listened to no sound for 30 min. The application of headphones and binaural beats was performed by a nurse who was not involved in the investigation. This study was a randomized controlled trial with blinding of anesthesiologists and outcome assessors. All participants wore identical headphones; however, because only the binaural-beats group received audible auditory stimulation, complete blinding of participants could not be ensured.

## Anesthetic procedure

The patients were transferred to the operating room, where they were monitored using non-invasive blood pressure measurement, electrocardiography, and pulse oximetry. The depth of anesthesia was assessed using the Patient State Index (PSI), which was measured using a SedLine® brain function monitor (Masimo, Irvine, CA, USA).

After preoxygenation with 100% oxygen, remimazolam was infused continuously at a rate of 6 mg/kg/h. LoC during anesthesia induction (LoC) was defined as the absence of response to standardized verbal commands. During remimazolam infusion, verbal commands ("Please open your eyes") were delivered every 5 seconds by the attending anesthesiologist using a predefined script. LoC was determined as the first time point at which the patient failed to respond to two consecutive commands. This protocol was applied consistently across all participants to ensure standardized assessment of LoC. Following LoC, continuous infusion of remifentanil was initiated with a target concentration of 4 ng/mL effect site concentration using the Minto pharmacodynamic model (Agilia SP TIVA; Fresenius Kabi, Bad Homburg, Germany) [17]. Additionally, rocuronium was administered at a dosage of 0.8 mg/kg. Tracheal intubation was performed after confirmation of complete muscle relaxation. Anesthesia was maintained using sevoflurane, along with a continuous infusion of remifentanil and rocuronium. Hypotension was defined as a mean arterial pressure of < 65 mmHg or a decrease in mean arterial pressure by 20% from baseline. During the first 30 min after anesthesia induction, hypotension was managed with vasopressors or inotropes at the discretion of the attending anesthesiologist. The incidence of hypotension requiring vasopressor or inotrope treatment was recorded during the first 30 min after anesthesia induction.

## Assessment and data collection

The anxiety score (0, no anxiety; 10, maximum anxiety) was assessed before and after the 30-min headphone application period. Anxiety was not reassessed after transfer to the operating room immediately before anesthesia induction. The remimazolam dose and the time to events, including the absence of response to vocal stimuli or the eyelash reflex and PSI ≤ 50, were evaluated. The time to LoC was defined as the time taken to achieve the absence of response to vocal stimuli. Hemodynamic variables during these events, as well as when the PSI dropped to ≤50, were also assessed. The PSI and relative power of electroencephalography (EEG) were recorded using a SedLine® brain function monitor. PSI monitoring was initiated at the start of anesthesia induction and was not recorded before headphone application or immediately before induction. Data on hemodynamic variables, PSI, spectral edge frequency (SEF), and relative power of the EEG were collected and analyzed using open-source VitalRecorder software (version 1.13.9) [18]. Relative EEG power

was calculated as the percentage of power in each frequency band (alpha: 8–12 Hz, beta: 12–30 Hz, delta: 0.5–4 Hz, gamma: 30–100 Hz, theta: 4–8 Hz) relative to the total EEG power, averaged from 10-s intervals recorded by the Sed-Line® monitor and analyzed using VitalRecorder software (version 1.13.9). Measurements were collected for 10 min before anesthesia induction and from the start of remimazolam infusion until LoC, with normalization based on the total power during each 10-s interval. Changes in relative EEG power were calculated as the post-anesthesia induction values minus the pre-anesthesia induction values for each group (binaural beats [B] group and control group). Positive values indicated an increase in relative power post-induction compared with pre-induction, while negative values indicated a decrease. Between-group differences in these changes (B group value minus control group value) were analyzed and reported with 95% confidence intervals and p-values.

## Statistical analyses

The primary outcome was the remimazolam dose required to achieve an absence of response to vocal stimuli. Based on a preliminary, unpublished pilot investigation, the remimazolam dose necessary for the absence of vocal stimuli was 17.8±5.1 mg. Assuming that the requirement would be 20% lower in the binaural beats group than in the control group, a sample size of 36 patients per group was necessary to achieve 80% power, with a two-sided significance level of 0.05 and a dropout rate of 10%. This assumed effect size was informed by the pilot data and supported by previous studies reporting reduced anesthetic requirements with binaural beat stimulation [9,10]. This 20% reduction was chosen as a conservative estimate for sample size calculation to achieve 80% power with a two-sided significance level of 0.05 and a 10% dropout rate.

For outcomes involving repeated measurements, a two-way repeated-measures analysis of variance (ANOVA) was used, with time (before vs. after anesthesia induction) as a within-subject factor and group (binaural beats vs. control) as a between-subject factor. When significant main effects or interactions were identified, post hoc comparisons were performed based on the ANOVA model. Assumptions of normality were evaluated using the residuals of the model. For secondary outcomes, p-values were interpreted cautiously with consideration of multiple comparisons. Continuous variables were presented as the mean±standard deviation. Categorical variables were compared using the chi-squared test or Fisher's exact test. All analyses were conducted on an intention-to-treat basis. Statistical significance was set at p < 0.05. Statistical analyses were performed using SPSS (version 25; IBM, Armonk, NY, USA) and R software (version 3.6.1; R Foundation for Statistical Computing, Vienna, Austria).

The detailed study protocol is provided in S1 and S2 File. The anonymized dataset used for the analysis is available in S1 Table. The study was reported in accordance with the CONSORT 2010 guidelines (S3 File).

## Results

Seventy-five patients were screened for inclusion in this study. Two patients were excluded owing to hearing disorders and one refused to participate. Therefore, 72 patients were included in the final analysis (Fig 1).

The demographic characteristics of patients in the B and control groups were comparable. No significant differences were observed in age, sex distribution, height, weight, body mass index, or ASA physical status between the two groups (Table 1). The anxiety scores before headphone application were similar between groups, but after headphone application, they were significantly lower in the B group than in the control group (3.0±2.8 vs. 4.4±2.5, p = 0.034).

The dose of remimazolam required to achieve the absence of response to vocal stimuli was significantly lower in the B group than in the control group (15.0±3.6 vs. 17.7±4.5 mg, p = 0.006, Table 2). Similarly, the dose of remimazolam per unit bodyweight was significantly lower in the B group than in the control group (0.23±0.05 vs. 0.31±0.18 mg/kg, p = 0.014). The time to the absence of response to vocal stimuli was also shorter in the B group than in the control group (140±29 vs. 168±47 s, p = 0.003).

 

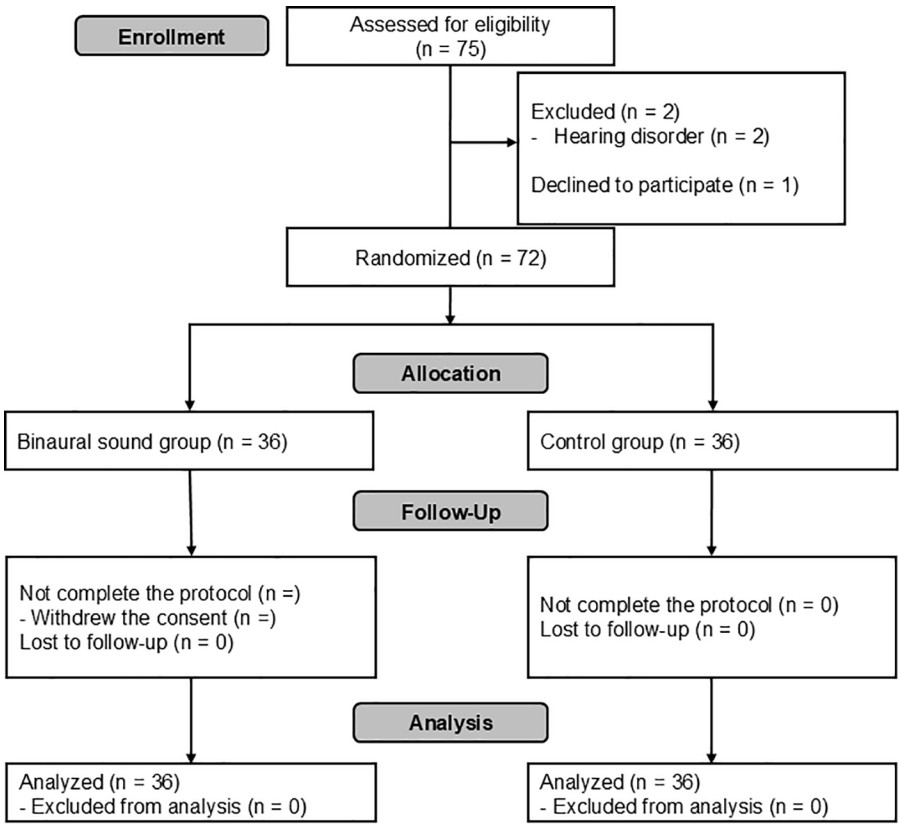

**Fig 1. CONSORT diagram.**

For the absence of eyelash reflex, the remimazolam dose was lower in the B group than in the control group (15.5 ± 3.7 vs. 18.4 ± 4.7 mg, p = 0.005), as was the dose per unit bodyweight (0.24 ± 0.05 vs. 0.29 ± 0.07 mg/kg, p = 0.001). The time to absence of eyelash reflex was shorter in the B group than in the control group (146 ± 34 vs. 174 ± 45 s, p = 0.003), and the PSI was higher (73 ± 16 vs. 62 ± 15, p = 0.003). No significant between-group differences were observed in the SEFs, mean blood pressure, or heart rate at this timepoint.

When the PSI was ≤ 50, no significant difference was observed in the remimazolam dose between the two groups (21.2 ± 6.1 vs. 22.2 ± 5.2 mg, p = 0.451), although the dose per unit bodyweight tended to be lower in the B group than in the control group (0.32 ± 0.08 vs. 0.36 ± 0.10 mg/kg, p = 0.142). No significant differences were observed in the SEFs, mean blood pressure, or heart rate between the groups at this timepoint.

During tracheal intubation, the PSI and hemodynamic variables (mean blood pressure and heart rate) did not differ significantly between the groups.

Two-way repeated-measures ANOVA was conducted to evaluate the effects of time (before vs. after anesthesia induction) and group (binaural beats vs. control) on relative EEG power across alpha, beta, delta, gamma, and theta bands.

No significant interaction was observed between time and group for any frequency band. The main effect of time was significant for all frequency bands, indicating changes in relative EEG power after anesthesia induction. The main effect of group was not significant for any frequency band. Post-hoc t-tests revealed no significant between-group differences in relative power before or after anesthesia induction (Table 3). Before anesthesia induction, the relative powers of the alpha, beta, delta, gamma, and theta waves did not differ significantly between the B and control groups. The relative power of

**Table 1. Demographic characteristics.**

|  | B group (n = 36) | Control group (n = 36) |
|---|---|---|
| Age | 47 ± 10 | 46 ± 12 |
| Sex |  |  |
| Male, % | 13 (36%) | 12 (33%) |
| Height, cm | 164 ± 7 | 164 ± 9 |
| Weight, kg | 66 ± 12 | 65 ± 14 |
| Body mass index, kg/m² | 24 ± 4 | 24 ± 3 |
| ASA physical status, 1/2 | 30 (83%)/6 (17%) | 30 (83%)/6 (17%) |
| Smoking, % | 3 (8%) | 3 (8%) |
| Anxiety score |  |  |
| Before headphone application | 4.7 ± 2.7 | 4.7 ± 2.7 |
| After headphone application | 3.0 ± 2.8 | 4.4 ± 2.5 |
| Surgery type |  |  |
| Robotic prostatectomy | 4 (11%) | 5 (14%) |
| Robotic cholecystectomy | 6 (17%) | 5 (14%) |
| Robotic myomectomy | 3 (8%) | 6 (17%) |
| Robotic gastrectomy | 5 (14%) | 3 (8%) |
| Robotic ovarian cystectomy | 7 (19%) | 6 (17%) |
| Robotic nephrectomy | 4 (11%) | 1 (3%) |
| Others | 7 (19%) | 10 (28%) |
| Anesthesia time | 174 ± 73 | 182 ± 76 |

B, binaural sound; ASA, American Society of Anesthesiologists. The B group listened to binaural sounds through headphones prior to anesthesia induction. The control group listened to no sound through headphones prior to anesthesia induction. The anxiety score was measured on a scale from 0 (no anxiety) to 10 (maximum imaginable anxiety).

these waves also did not differ significantly between groups after anesthesia induction. However, changes in the relative power between before and after anesthesia induction were significant for all wave types. The relative power difference between before and after anesthesia induction did not differ significantly between groups.

The incidence of hypotension was significantly lower in the B group than in the control group (2 [6%] vs. 10 [28%], p = 0.024). The incidence of vasopressor use was lower in the B group than in the control group (2 [6%] vs. 10 [28%], p = 0.024).

## Discussion

This study investigated the effects of preoperative binaural beats on the remimazolam dose required for LoC (absence of response to vocal stimuli) during general anesthesia induction. Our results demonstrated that the preoperative application of binaural beats significantly reduced the dose of remimazolam required for LoC, as assessed by the response to vocal stimuli and the eyelash reflex. In addition, binaural beats reduced the incidence of hypotension during anesthesia induction.

Previous studies have demonstrated that binaural beats can reduce the requirements for anesthetics and analgesics [9,10,12]. Our study extends these findings by being the first to investigate the effects of preoperative binaural beats on patients being administered remimazolam, a novel ultra-short-acting benzodiazepine with rapid onset, offset, and favorable hemodynamic stability [1–3]. This is particularly significant given the limited data on optimizing

**Table 2. Variables of anesthesia induction using remimazolam infusion.**

| | B group (n = 36) | Control group (n = 36) | Difference (95% confidence interval) | p-value |
|---|---|---|---|---|
| Loss of consciousness (defined as absence of response to vocal stimuli) | | | | |
| **Remimazolam dose, mg** | 15.0±3.6 | 17.7±4.5 | −2.7 (−4.6 to −0.8) | 0.006 |
| **Remimazolam dose per unit bodyweight, mg/kg** | 0.23±0.05 | 0.31±0.18 | −0.07 (−0.14 to −0.02) | 0.014 |
| **Duration, s** | 140±29 | 168±47 | −28 (−47 to −10) | 0.003 |
| **Patient state index** | 76±16 | 63±16 | 12 (5–20) | 0.002 |
| **Right spectral edge frequency** | 14±6 | 15±6 | −1 (−4–2) | 0.534 |
| **Left spectral edge frequency** | 14±6 | 14±5 | 0 (−3–2) | 0.910 |
| **Mean blood pressure, mmHg** | 90±19 | 88±19 | 1 (−8–10) | 0.785 |
| **Heart rate, bpm** | 79±11 | 81±13 | −3 (−8–3) | 0.372 |
| In the absence of the eyelash reflex | | | | |
| **Remimazolam dose, mg** | 15.5±3.7 | 18.4±4.7 | −2.6 (−4.9 to −0.9) | 0.005 |
| **Remimazolam dose per unit bodyweight, mg/kg** | 0.24±0.05 | 0.29±0.07 | −0.05 (−0.08 to −0.02) | 0.001 |
| **Duration, s** | 146±34 | 174±45 | −29 (−47 to −10) | 0.003 |
| **Patient state index** | 73±16 | 62±15 | 12 (4–19) | 0.003 |
| **Right spectral edge frequency** | 14±6 | 15±6 | −1 (−4–2) | 0.530 |
| **Left spectral edge frequency** | 14±6 | 15±5 | 0 (−3–2) | 0.730 |
| **Mean blood pressure, mmHg** | 89±19 | 87±17 | 2 (−7–10) | 0.645 |
| **Heart rate, bpm** | 79±11 | 81±14 | −3 (−8–3) | 0.395 |
| In the patient state index ≤ 50 | | | | |
| **Remimazolam dose, mg** | 21.2±6.1 | 22.2±5.2 | −1.0 (−3.7 to 1.6) | 0.451 |
| **Remimazolam dose per unit bodyweight, mg/kg** | 0.32±0.08 | 0.36±0.10 | −0.05 (−0.08 to −0.02) | 0.142 |
| **Duration, s** | 228±62 | 245±101 | −17 (−56–22) | 0.389 |
| **Right spectral edge frequency** | 13±5 | 15±4 | −2 (−4–0) | 0.083 |
| **Left spectral edge frequency** | 13±6 | 15±4 | −2 (−5–0) | 0.089 |
| **Mean blood pressure, mmHg** | 89±16 | 90±15 | −2 (−9–5) | 0.618 |
| **Heart rate, bpm** | 77±13 | 82±14 | −5 (−11–2) | 0.137 |
| At tracheal intubation | | | | |
| **Patient state index** | 42±7 | 45±7 | −3 (−6–0) | 0.079 |
| **Mean blood pressure, mmHg** | 106±27 | 104±23 | 2 (−10–14) | 0.752 |
| **Heart rate, bpm** | 92±11 | 95±13 | −3 (−9–3) | 0.307 |

B, binaural sound. The B group listened to binaural sounds through headphones prior to anesthesia induction. The control group listened to no sound through headphones prior to anesthesia induction.

remimazolam dosing during anesthesia induction, where individual variability and potential side effects like hypotension remain concerns [4–7]. By demonstrating that binaural beats reduce the remimazolam dose required for LoC and the incidence of hypotension, our study introduces a non-invasive adjunct that enhances the safety and efficiency of remimazolam-based anesthesia induction. Furthermore, while our EEG spectral analysis did not detect significant between-group differences in standard frequency bands (alpha, beta, delta, gamma, theta), the use of a 1-Hz binaural beat frequency suggests the potential for frequency-specific neural entrainment at 1 Hz, which was not explored in this study due to the focus on broader frequency bands [15]. Future investigations could incorporate targeted 1-Hz power analysis to assess whether binaural beats induce specific neural responses that further explain the observed reductions in remimazolam requirements, offering a novel avenue of examining brainwave entrainment mechanisms in anesthesia.

**Table 3. Electroencephalography during anesthesia induction.**

| | B group (n = 36) | Control group (n = 36) | Difference (95% confidence interval) | p-value |
|---|---|---|---|---|
| Relative power before anesthesia induction, % | | | | |
| **Alpha** | 3 ± 3 | 4 ± 5 | −1 (−3–1) | 0.346 |
| **Beta** | 4 ± 4 | 5 ± 5 | −1 (−3–2) | 0.614 |
| **Delta** | 77 ± 10 | 75 ± 12 | 2 (−4–7) | 0.555 |
| **Gamma** | 2 ± 2 | 2 ± 2 | 0 (−1–1) | 0.931 |
| **Theta** | 10 ± 3 | 10 ± 4 | 0 (−2–2) | 0.954 |
| Relative power after anesthesia induction, % | | | | |
| **Alpha** | 8 ± 8 | 8 ± 8 | −1 (−4–4) | 0.956 |
| **Beta** | 17 ± 16 | 18 ± 17 | 0 (−8–7) | 0.897 |
| **Delta** | 57 ± 25 | 58 ± 25 | 0 (−12–12) | 0.965 |
| **Gamma** | 3 ± 2 | 2 ± 2 | 0 (−1–1) | 0.546 |
| **Theta** | 9 ± 3 | 9 ± 3 | 0 (−1–2) | 0.678 |
| Relative power difference before and after anesthesia induction | | | | |
| **Alpha** | 5 ± 7 | 4 ± 8 | −3–5 | 0.589 |
| **Beta** | 13 ± 17 | 13 ± 18 | −8–9 | 0.896 |
| **Delta** | −19 ± 25 | −19 ± 28 | −13–12 | 0.969 |
| **Gamma** | 1 ± 3 | 1 ± 3 | −1–2 | 0.525 |
| **Theta** | −1 ± 3 | −1 ± 4 | −1–2 | 0.522 |

B, binaural sound. The B group listened to binaural sounds through headphones prior to anesthesia induction. The control group listened to no sound through headphones prior to anesthesia induction.

A practical challenge in using binaural beats during anesthesia is the necessity for patients to wear headphones, which can be difficult if they are not supine. However, the application of binaural beats in the preoperative treatment room is relatively straightforward and practical. In our study, the administration of binaural beats before anesthesia induction effectively reduced the required dose of remimazolam. Additionally, a reduction in the time to LoC of approximately 28 s could be clinically significant. Given that heightened anxiety during induction can prolong the process and adversely affect postoperative outcomes, including quality of life [8,19], this shorter induction time, combined with reduced preoperative anxiety, may contribute to a less stressful experience for patients. Furthermore, the preoperative use of binaural beats presents a non-invasive, user-friendly intervention that could improve patient outcomes by enhancing the efficiency and safety of anesthesia induction. This approach aligns with ongoing efforts in anesthesiology to minimize drug doses and enhance patient comfort and safety.

Remimazolam is widely recognized for its safety profile, particularly its hemodynamic stability [2,3]. However, when administered at high doses, it can lead to significant hypotension, necessitating the use of vasopressors to maintain hemodynamic stability [1]. In our study, the preoperative application of binaural beats significantly reduced the incidence of hypotension requiring vasopressor intervention during anesthesia induction. This reduction is likely attributable to the reduced dosage of remimazolam required when binaural beats are employed, as they have previously been shown to enhance the efficacy of anesthesia agents, allowing for lower dosages. Vasopressors, which are effective in managing hypotension, are associated with risks and complications. Their use can lead to adverse effects, such as tachycardia, myocardial ischemia, and peripheral vasoconstriction, which pose significant risks, particularly in patients with pre-existing cardiovascular conditions [20]. Thus, minimizing the need for vasopressors through adjunctive measures such as binaural beats can be highly beneficial. Reduced hypotension may lessen the need for vasopressors and their associated risks, thereby enhancing overall safety and workflow efficiency. From this perspective, the incorporation of preoperative

binaural beats into anesthetic protocols holds promise for the enhancement of the safety of anesthesia induction using remimazolam.

The PSI at the time of LoC differed significantly between the two groups, despite a shorter time to LoC and lower remimazolam dose in the binaural-beats group. This finding should be interpreted with caution, as PSI values are calculated using processed EEG data with inherent smoothing and display delays. As a result, the PSI value displayed at the behavioral endpoint of loss of consciousness may not precisely reflect the instantaneous cortical state at that moment, particularly when loss of consciousness occurs rapidly. This temporal mismatch may partly explain the observed between-group difference in PSI at the time of loss of consciousness.

EEG was recorded using the SedLine® monitor, which derives spectral indices from a limited frontal montage; therefore, the EEG measures represent indirect assessments of frontal cortical activity and may not capture region-specific or distributed neural entrainment effects. In addition, this study was powered for clinical outcomes rather than mechanistic EEG endpoints, and subtle between-group differences in EEG spectral power may have been underdetected. Our analysis, which focused on standard frequency bands (alpha, beta, delta, gamma, theta), did not reveal significant between-group differences in EEG spectral power. Given this, the term "brainwave modulation", which encompasses broader changes in neural activity or relaxation states without requiring specific frequency synchronization, may more accurately describe the potential effects of binaural beats in our study. The lack of observed EEG changes could be attributed to the brief 30-min duration of binaural beat application, the focus on broad frequency bands rather than targeted 1-Hz power analysis, or insufficient statistical power to detect subtle variations [15]. While our EEG spectral analysis did not detect significant between-group differences in standard frequency bands (alpha, beta, delta, gamma, theta), a targeted analysis at the 1-Hz stimulation frequency could provide critical evidence for the hypothesized brainwave entrainment mechanism. Such an analysis was not performed in this study due to limitations in the EEG analysis configuration, which focused on broader frequency bands. Future studies should incorporate narrow-band spectral analysis or time-frequency decomposition targeting the 1-Hz frequency to determine whether binaural beats induce specific neural entrainment, potentially transforming our understanding of their role in optimizing anesthesia induction. This could bridge a critical gap in the literature and establish a mechanistic foundation for the use of binaural beats in anesthesiology. It is also possible that binaural beats primarily reduced remimazolam requirements through anxiety reduction, rather than direct neural entrainment. Future studies should extend the duration of binaural beat exposure and include frequency-specific analyses, such as 1-Hz power, to better elucidate the mechanisms by which binaural beats influence anesthetic requirements. The lower anxiety scores in the binaural beats group suggest that anxiety reduction may reduce the required remimazolam dosage. A post-hoc correlation analysis (e.g., Pearson's or Spearman's) between post-headphone anxiety scores and the remimazolam dose could confirm this but was not conducted in this study. Future studies should explore this using our shared dataset (Mendeley Data, DOI: 10.17632/swxgj79jpy.2) to clarify the role of anxiety reduction in optimizing anesthetic dosing. The absence of significant EEG changes in our study suggests that brainwave entrainment may not be the primary mechanism by which binaural beats reduce remimazolam requirements. This could be due to the brief 30-min exposure to 1-Hz binaural beats, which may be insufficient to induce detectable neural entrainment, or the lack of frequency-specific analysis targeting the 1-Hz stimulation frequency. Alternatively, the observed reduction in anxiety scores in the B group suggests that anxiety reduction may play a significant role in decreasing the required remimazolam dosage and time to LoC. Anxiety is known to increase anesthetic requirements [8], and correlation of anxiety scores with remimazolam dosage in future studies could clarify this mechanism. Low-frequency binaural beats, such as the 1-Hz difference used in this study, have been shown to promote deep sleep or reduced consciousness [14,21]. It is possible that preoperative binaural beat exposure induced a sleep-like state, lowering the baseline level of consciousness and thus reducing the remimazolam dose required for LoC. This hypothesis warrants further investigation using objective sedation measures, such as the Ramsay Sedation Scale or polysomnography, to assess the depth of relaxation during the preoperative period. Future studies could consider delivering binaural beats during the anesthesia induction phase to allow a more direct assessment of neural entrainment during the transition to unconsciousness.

LoC in the present study was assessed using standardized verbal commands delivered at fixed 5-second intervals, which allowed for a practical and reproducible determination of unresponsiveness in the operating room setting. Alternative approaches, such as switch-based paradigms in which patients actively confirm consciousness at predefined intervals, have been used in experimental settings to achieve higher temporal resolution. While such methods may offer more precise detection of the transition to unconsciousness, their application during routine clinical anesthesia induction may be limited by feasibility and workflow considerations.

The current study had some limitations. First, this was a single-center study with a relatively small sample size, which may limit the generalizability of the findings to other institutions, patient populations, or perioperative settings. Multicenter studies with larger cohorts are needed to confirm the robustness of these results. Second, EEG analysis was based on a limited frontal montage and broad frequency bands, and the study was not powered to detect subtle or frequency-specific neuralentrainment effects. In particular, narrow-band analysis targeting the 1-Hz stimulation frequency was not performed, which may have limited mechanistic interpretation of the EEG findings. Third, although anxiety scores were reduced in the binaural-beats group, anxiety reduction and neural entrainment could not be disentangled mechanistically, as no correlation, mediation, or causal pathway analyses were performed. Future studies incorporating formal mediation models may help clarify the relative contributions of psychological and neurophysiological mechanisms. Finally, the findings are specific to remimazolam-based anesthesia induction and may not be directly generalizable to other anesthetic agents with different pharmacodynamic and hemodynamic profiles.

## Conclusions

Our study suggests that the preoperative use of binaural beats may reduce the dose of remimazolam needed for LoC (the absence of response to vocal stimuli), shorten the time to achieve it, and potentially lower the incidence of hypotension during anesthesia induction. These preliminary findings indicate that binaural beats could serve as a non-invasive adjunctive measure to support the efficiency and safety of anesthesia induction, though further research with larger samples is needed to confirm these effects.

## Supporting information

**S1 File. Study protocol in English.**
(DOCX)

**S2 File. Study protocol in Korean.**
(DOCX)

**S1 Table. Raw anonymized dataset.**
(XLSX)

**S3 File. CONSORT 2010 Checklist.**
(DOCX)

**S4 File. Graphic abstract.**
(TIFF)

## Acknowledgments

We would like to thank Medical Illustration & Design (MID), a member of the Medical Research Support Services of Yonsei University College of Medicine, for providing excellent support with the medical illustrations. A graphical abstract is provided to summarize the study design and main findings (S4 File).

## Author contributions

**Conceptualization:** Hyun-Chang Kim, Jin Young Sohn, Myoung Hwa Kim, Yoon Jung Kim, Chul Ho Chang, Jeong-Hwa Seo.

**Data curation:** Hyun-Chang Kim, Jeong-Hwa Seo.

**Formal analysis:** Hyun-Chang Kim, Jeong-Hwa Seo.

**Funding acquisition:** Hyun-Chang Kim, Jeong-Hwa Seo.

**Investigation:** Hyun-Chang Kim, Jin Young Sohn, Myoung Hwa Kim, Yoon Jung Kim, Chul Ho Chang, Jeong-Hwa Seo.

**Methodology:** Hyun-Chang Kim, Jin Young Sohn, Myoung Hwa Kim, Yoon Jung Kim, Chul Ho Chang, Jeong-Hwa Seo.

**Project administration:** Hyun-Chang Kim, Jeong-Hwa Seo.

**Resources:** Hyun-Chang Kim, Myoung Hwa Kim, Jeong-Hwa Seo.

**Software:** Hyun-Chang Kim, Jeong-Hwa Seo.

**Supervision:** Hyun-Chang Kim, Jin Young Sohn, Yoon Jung Kim, Chul Ho Chang, Jeong-Hwa Seo.

**Validation:** Hyun-Chang Kim, Jeong-Hwa Seo.

**Visualization:** Hyun-Chang Kim, Jin Young Sohn, Myoung Hwa Kim, Yoon Jung Kim, Chul Ho Chang, Jeong-Hwa Seo.

**Writing – original draft:** Hyun-Chang Kim, Jin Young Sohn, Myoung Hwa Kim, Yoon Jung Kim, Chul Ho Chang, Jeong-Hwa Seo.

**Writing – review & editing:** Hyun-Chang Kim, Jin Young Sohn, Myoung Hwa Kim, Yoon Jung Kim, Chul Ho Chang, Jeong-Hwa Seo.

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
