## [Decision Letter · Decision Letter 0]

4 Feb 2026

Dear Dr. Seo,

Thank you for submitting your manuscript to PLOS ONE. After careful consideration, we feel that it has merit but does not fully meet PLOS ONE’s publication criteria as it currently stands. Therefore, we invite you to submit a revised version of the manuscript that addresses the points raised during the review process.

We look forward to receiving your revised manuscript.

Kind regards,

Nabin Lageju

Guest Editor

PLOS One

Journal Requirements:

https://journals.plos.org/plosone/s/file?id=wjVg/PLOSOne_formatting_sample_main_body.pdf and and and and https://journals.plos.org/plosone/s/file?id=ba62/PLOSOne_formatting_sample_title_authors_affiliations.pdf

This work was supported by the Department of Anesthesiology and Pain Medicine and Anesthesia and Pain Research Institute, Yonsei University College of Medicine. This research was supported by a special research grant funded by the Korean Society of Neuroscience in Anesthesiology and Critical Care (KSNACC-2024) and a new faculty research seed money grant from the Yonsei University College of Medicine for 2024 (2024-32-0075).

4. Please remove all personal information, ensure that the data shared are in accordance with participant consent, and re-upload a fully anonymized data set.

Additional Editor Comments:

Please follow the comments and submit for revision.

Reviewers' comments:

Reviewer's Responses to Questions

**Comments to the Author**

1. Is the manuscript technically sound, and do the data support the conclusions?

Reviewer #1: Yes

Reviewer #2: Yes

Reviewer #3: Yes

2. Has the statistical analysis been performed appropriately and rigorously?

Reviewer #1: No

Reviewer #2: Yes

Reviewer #3: Yes

3. Have the authors made all data underlying the findings in their manuscript fully available?

Reviewer #1: Yes

Reviewer #2: Yes

Reviewer #3: Yes

4. Is the manuscript presented in an intelligible fashion and written in standard English?

Reviewer #1: Yes

Reviewer #2: Yes

Reviewer #3: Yes

Reviewer #1: Lines 124 and 128 contain the same information and should be revised.

Line 133: For repeated measurements, repeated-measures ANOVA or mixed-effects models should be used.

Line 135: Post hoc tests should be based on the primary analysis model. Normality should be assessed using the residuals of this model. The Mann–Whitney U test should not be used as a post hoc test.

Remove Cohen’s d from all tables.

P values for secondary outcomes need to be adjusted for multiple comparisons.

Line 182: no need to report F values or partial eta-squared values.

Reviewer #2: This study provides valuable clinical insights by demonstrating that preoperative binaural beats can significantly reduce the required dosage of remimazolam during anesthesia induction. It effectively introduces a non-invasive adjunct that not only optimizes anesthetic efficiency but also enhances patient safety by reducing preoperative anxiety and the incidence of hypotension. While the manuscript is well-structured and addresses a compelling topic, I would like to offer the following comments for further clarification and improvement:

1. The authors state that the specific frequency difference of 1 Hz for the binaural beats was selected based on reference 14. However, reference 14 focuses on how propofol anesthesia alters cortical traveling waves and does not appear to provide a specific rationale or evidence for the selection of a 1 Hz binaural beat frequency. I recommend the authors provide a more accurate citation or a clearer scientific justification for why this specific frequency was chosen for the intervention.

2. The primary parameter in this study is the remimazolam dose required to achieve loss of consciousness, which was defined as the absence of response to vocal stimuli. To ensure the robustness of this data, a more detailed explanation of the assessment protocol is required. Specifically, it is important to clarify the frequency and content of the verbal commands used, as long intervals between commands can make it difficult to accurately detect the precise moment of loss of consciousness. The authors might consider comparing their protocol to other established methods, such as utilizing prompts every 5 seconds where patients confirm consciousness by pressing a switch.

3. The study analyzed EEG spectral power but found no significant differences between the groups after anesthesia induction. This finding suggests a need for further discussion regarding whether preoperative binaural beats can induce lasting modifications in brainwave patterns that persist after the onset of general anesthesia. Additionally, the authors could consider whether delivering the binaural beats during the induction phase, rather than only preoperatively, would provide a more direct measurement of neural entrainment and its impact on EEG variables.

Reviewer #3: This is a well-designed, single-center, randomized controlled trial whether 30 minutes of preoperative 1-Hz binaural beats reduce the total remimazolam dose and time required to achieve loss of consciousness (LoC; unresponsiveness to verbal stimuli) during induction with continuous remimazolam infusion (6 mg/kg/h), and whether they decrease the incidence of hypotension (or vasopressor use) within 30 minutes after induction. The main findings were that the binaural-beats group required a lower remimazolam dose and a shorter time to reach LoC, and experienced fewer hypotensive events.

I have a few comments.

1. Although the study is described as double-blind, it is unlikely that participants were truly blinded. The authors should clearly specify who was blinded (e.g., participants, anesthesiologists, outcome assessors) and how blinding was maintained.

2. Because the headphone (binaural beats) group reached LoC more quickly, they received a lower total dose of remimazolam. However, PSI at the time of LoC differed between the two groups, which is not intuitively straightforward. To aid readers’ interpretation, the authors should add a discussion addressing possible explanations for this discrepancy.

3. Are there anxiety and/or PSI values measured (1) before and after headphone use, and (2) immediately before induction (after transfer to the operating room)?

4. There are more than ten limitations listed. The authors should consider removing limitations that are not directly relevant to this study, and incorporating any essential points into a more focused discussion instead.

For example, in my opinion, the second and third limitations do not appear to be direct limitations of the present study. The seventh and eighth points relate more to study design choices and would be better addressed in the Methods and/or Discussion rather than framed as limitations. In particular, the authors should explain why a 1-Hz frequency difference was chosen and why the control condition was silence rather than a 0-Hz (sham) auditory stimulus.

5. I would like to know whether the authors assessed changes in anxiety scores among participants who wore headphones with no audible sound for 30 minutes.

.

Reviewer #1: No

Reviewer #2: No

Reviewer #3: No

---

## [Author Response · Author response to Decision Letter 1]

10 Feb 2026

Journal Requirements:

Response:

We have carefully reviewed and revised the manuscript to ensure full compliance with PLOS ONE’s style requirements.

The title page and main manuscript were formatted according to the official PLOS ONE templates, including section organization, headings, and file naming conventions.

This work was supported by the Department of Anesthesiology and Pain Medicine and Anesthesia and Pain Research Institute, Yonsei University College of Medicine. This research was supported by a special research grant funded by the Korean Society of Neuroscience in Anesthesiology and Critical Care (KSNACC-2024) and a new faculty research seed money grant from the Yonsei University College of Medicine for 2024 (2024-32-0075).

Response:

We have included the following Role of the Funder statement in the cover letter, as requested:

Response:

We have added a Supporting Information section at the end of the manuscript, including captions for all Supporting Information files, and updated the in-text citations to match accordingly.

4. Please remove all personal information, ensure that the data shared are in accordance with participant consent, and re-upload a fully anonymized data set.

Response:

We have rechecked the dataset and confirmed that all personally identifiable information, including all date-related variables, has been completely removed.

The re-uploaded dataset is fully anonymized and contains only non-identifiable variables in accordance with participant consent and the PLOS Data Policy.

Response:

We reviewed the reviewer comments and evaluated whether any additional previously published works were recommended for citation.

As no specific references were suggested by the reviewers, no changes were made to the reference list.

Additional Editor Comments:

Please follow the comments and submit for revision.

Reviewers' comments:

Reviewer's Responses to Questions

Comments to the Author

5. Review Comments to the Author

Reviewer #1: Lines 124 and 128 contain the same information and should be revised.

Response:

We revised the sentence describing the basis for the assumed 20% effect size.

The original detailed description was replaced with a concise statement indicating that the effect size was informed by pilot data and supported by previous literature.

Line 133: For repeated measurements, repeated-measures ANOVA or mixed-effects models should be used.

Response:

In response to the reviewer’s comment, we have revised the Statistical Analysis section to explicitly state that repeated-measures ANOVA was applied for repeated measurements, rather than simple pairwise tests.

Line 135: Post hoc tests should be based on the primary analysis model. Normality should be assessed using the residuals of this model. The Mann–Whitney U test should not be used as a post hoc test.

Response: We agree with the reviewer’s comment.

The Statistical Analysis section has been revised to clarify that post hoc comparisons were conducted based on the primary repeated-measures ANOVA model, and that normality assumptions were assessed using the residuals of the model.

In addition, inappropriate references to the Mann–Whitney U test as a post hoc procedure have been removed.

Remove Cohen’s d from all tables.

Response: Cohen’s d values have been removed from all tables as recommended.

P values for secondary outcomes need to be adjusted for multiple comparisons.

Response:

We agree with the reviewer’s comment.

The Statistical Analysis section has been revised to clarify that p-values for secondary outcomes were interpreted cautiously with consideration of multiple comparisons.

Line 182: no need to report F values or partial eta-squared values.

Response:

We agree with the reviewer’s comment.

The Results section has been revised to remove the reporting of F statistics and partial eta-squared values, and the findings are now described more concisely.

Reviewer #2: This study provides valuable clinical insights by demonstrating that preoperative binaural beats can significantly reduce the required dosage of remimazolam during anesthesia induction. It effectively introduces a non-invasive adjunct that not only optimizes anesthetic efficiency but also enhances patient safety by reducing preoperative anxiety and the incidence of hypotension. While the manuscript is well-structured and addresses a compelling topic, I would like to offer the following comments for further clarification and improvement:

Response:

We thank the reviewer for the positive and constructive overall assessment of our study.

1. The authors state that the specific frequency difference of 1 Hz for the binaural beats was selected based on reference 14. However, reference 14 focuses on how propofol anesthesia alters cortical traveling waves and does not appear to provide a specific rationale or evidence for the selection of a 1 Hz binaural beat frequency. I recommend the authors provide a more accurate citation or a clearer scientific justification for why this specific frequency was chosen for the intervention.

Response:

We appreciate the reviewer’s insightful comment. We agree that reference 14 alone does not directly justify the selection of a 1-Hz binaural beat frequency. Accordingly, we have revised the manuscript to clarify the scientific rationale for this choice and added references demonstrating that very low-frequency binaural beats are associated with

sleep promotion and reduced levels of consciousness. Reference 14 is now cited as supporting evidence for the relevance of slow oscillations during anesthesia, rather than as a direct justification for the binaural beat frequency selection.

2. The primary parameter in this study is the remimazolam dose required to achieve loss of consciousness, which was defined as the absence of response to vocal stimuli. To ensure the robustness of this data, a more detailed explanation of the assessment protocol is required. Specifically, it is important to clarify the frequency and content of the verbal commands used, as long intervals between commands can make it difficult to accurately detect the precise moment of loss of consciousness. The authors might consider comparing their protocol to other established methods, such as utilizing prompts every 5 seconds where patients confirm consciousness by pressing a switch.

Response:

We thank the reviewer for this important comment.

Loss of consciousness was assessed using standardized verbal commands delivered every 5 seconds (“Please open your eyes”) during remimazolam infusion. LoC was defined as the absence of response to two consecutive commands, allowing for precise temporal

identification of the transition to unconsciousness. We have clarified this assessment protocol in the Methods section and the study protocol (S1 File).

We have added a discussion comparing our method for assessing loss of consciousness

with switch-based paradigms, as suggested by the reviewer.

3. The study analyzed EEG spectral power but found no significant differences between the groups after anesthesia induction. This finding suggests a need for further discussion regarding whether preoperative binaural beats can induce lasting modifications in brainwave patterns that persist after the onset of general anesthesia. Additionally, the authors could consider whether delivering the binaural beats during the induction phase, rather than only preoperatively, would provide a more direct measurement of neural entrainment and its impact on EEG variables.

Response:

We thank the reviewer for this important comment. The lack of significant EEG differences after anesthesia induction may reflect the dominant neurophysiological effects of general anesthesia, which could mask subtle preoperative entrainment effects. In addition, EEG was recorded using a limited frontal montage (SedLine®), and the study was powered for clinical rather than mechanistic EEG endpoints; therefore, subtle differences in spectral power may have been underdetected.

We agree that delivering binaural beats during the induction phase may provide a more direct assessment of neural entrainment. This point has been added to the Discussion as an important direction for future research.

Reviewer #3: This is a well-designed, single-center, randomized controlled trial whether 30 minutes of preoperative 1-Hz binaural beats reduce the total remimazolam dose and time required to achieve loss of consciousness (LoC; unresponsiveness to verbal stimuli) during induction with continuous remimazolam infusion (6 mg/kg/h), and whether they decrease the incidence of hypotension (or vasopressor use) within 30 minutes after induction. The main findings were that the binaural-beats group required a lower remimazolam dose and a shorter time to reach LoC, and experienced fewer hypotensive events.

Response:

We thank the reviewer for the accurate summary of our study and its main findings.

I have a few comments.

1. Although the study is described as double-blind, it is unlikely that participants were truly blinded. The authors should clearly specify who was blinded (e.g., participants, anesthesiologists, outcome assessors) and how blinding was maintained.

Response:

We thank the reviewer for this important point. Participants in both groups wore identical headphones; however, because only the intervention group received audible binaural beats, complete participant blinding cannot be guaranteed. We have clarified in the Methods section that anesthesiologists and outcome assessors were blinded to group allocation, and we have specified how blinding was maintained. The description of blinding has been revised accordingly.

2. Because the headphone (binaural beats) group reached LoC more quickly, they received a lower total dose of remimazolam. However, PSI at the time of LoC differed between the two groups, which is not intuitively straightforward. To aid readers’ interpretation, the authors should add a discussion addressing possible explanations for this discrepancy.

Response:

We thank the reviewer for this thoughtful comment. The observed difference in PSI values at the time of loss of consciousness may be related to the inherent processing delay and smoothing window used in PSI calculation. PSI values are derived from processed EEG data and may not represent instantaneous cortical states at the exact behavioral endpoint of unresponsiveness. Therefore, when loss of consciousness occurs rapidly, particularly in the binaural-beats group, a temporal mismatch between the clinical endpoint and the displayed PSI value may arise. We have added a clarification of this point in the Discussion.

3. Are there anxiety and/or PSI values measured (1) before and after headphone use, and (2) immediately before induction (after transfer to the operating room)?

Response:

We thank the reviewer for this question. Anxiety was assessed before and after the 30-minute headphone application period, as described in the Methods. However, anxiety was not reassessed after transfer to the operating room immediately before induction. PSI monitoring was initiated at the start of anesthesia induction and was not recorded either before or after headphone use, nor immediately before induction. We have clarified the timing of anxiety and PSI measurements in the Methods section.

4. There are more than ten limitations listed. The authors should consider removing limitations that are not directly relevant to this study, and incorporating any essential points into a more focused discussion instead.

For example, in my opinion, the second and third limitations do not appear to be direct limitations of the present study. The seventh and eighth points relate more to study design choices and would be better addressed in the Methods and/or Discussion rather than framed as limitations. In particular, the authors should explain why a 1-Hz frequency difference was chosen and why the control condition was silence rather than a 0-Hz (sham) auditory stimulus.

Response:

We agree with the reviewer that the original Limitations section was overly extensive

and included items that reflected study design choices rather than true limitations. Accordingly, we have substantially revised and condensed the Limitations section, removing points that were not directly related to methodological constraints and integrating essential considerations into the Methods and Discussion. In particular, the rationale for selecting a 1-Hz frequency difference and for using a silence control condition has been explicitly clarified in the Methods and Discussion, rather than being framed as limitations.

5. I would like to know whether the authors assessed changes in anxiety scores among participants who wore headphones with no audible sound for 30 minutes.

Response:

Yes. Anxiety scores were assessed both before and after the 30-minute headphone

application period in both groups. In the control group, anxiety scores did not significantly decrease after wearing headphones without audible sound, whereas a significant reduction was observed in the binaural-beats group. These data are presented in Table 1.

---

## [Decision Letter · Decision Letter 1]

12 Mar 2026

Preoperative binaural beats reduce remimazolam dosage and enhance safety in anesthesia induction: a randomized controlled trial

PONE-D-25-56587R1

Dear Dr. Seo,

We’re pleased to inform you that your manuscript has been judged scientifically suitable for publication and will be formally accepted for publication once it meets all outstanding technical requirements.

Kind regards,

Nabin Lageju

Guest Editor

PLOS One

Additional Editor Comments (optional):

Reviewers' comments:

Reviewer's Responses to Questions

**Comments to the Author**

Reviewer #1: All comments have been addressed

2. Is the manuscript technically sound, and do the data support the conclusions?

Reviewer #1: (No Response)

3. Has the statistical analysis been performed appropriately and rigorously?

Reviewer #1: (No Response)

4. Have the authors made all data underlying the findings in their manuscript fully available?

Reviewer #1: (No Response)

5. Is the manuscript presented in an intelligible fashion and written in standard English?

Reviewer #1: (No Response)

Reviewer #1: All my concerns are addressed.

.

Reviewer #1: No
